# Proteomic Profiling of the First Human Dental Pulp Mesenchymal Stem/Stromal Cells from Carbonic Anhydrase II Deficiency Osteopetrosis Patients

**DOI:** 10.3390/ijms22010380

**Published:** 2020-12-31

**Authors:** Zikra Alkhayal, Zakia Shinwari, Ameera Gaafar, Ayodele Alaiya

**Affiliations:** 1Therapeutics & Biomarker Discovery for Clinical Applications, Stem Cell & Tissue Re-Engineering Program, King Faisal Specialist Hospital and Research Centre, P.O. Box 3354, Riyadh 11211, Saudi Arabia; SZakia@kfshrc.edu.sa (Z.S.); agaafar@kfshrc.edu.sa (A.G.); 2Department of Dentistry, King Faisal Specialist Hospital & Research Center, P.O. Box 3354, Riyadh 11211, Saudi Arabia

**Keywords:** human-exfoliated-deciduous-teeth, mesenchymal stem/stromal cells (MSSCs), proteomics, osteopetrosis, carbonic anhydrase II, dental pulp

## Abstract

Osteopetrosis is a hereditary disorder characterized by sclerotic, thick, weak, and brittle bone. The biological behavior of mesenchymal cells obtained from osteopetrosis patients has not been well-studied. Isolated mesenchymal stem/stromal cells from dental pulp (DP-MSSCs) of recently extracted deciduous teeth from osteopetrosis (OP) patients and healthy controls (HCs) were compared. We evaluated whether the dental pulp of OP patients has a population of MSSCs with similar multilineage differentiation capability to DP-MSSCs of healthy subjects. Stem/progenitor cells were characterized using immunohistochemistry, flow cytometry, and proteomics. Our DP-MSSCs were strongly positive for CD44, CD73, CD105, and CD90. DP-MSSCs obtained from HC subjects and OP patients showed similar patterns of proliferation and differentiation as well as gene expression. Proteomic analysis identified 1499 unique proteins with 94.3% similarity in global protein fingerprints of HCs and OP patients. Interestingly, we observed subtle differences in expressed proteins of osteopetrosis disease-related in pathways, including MAPK, ERK 1/2, PI3K, and integrin, rather than in the stem cell signaling network. Our findings of similar protein expression signatures in DP-MSSCs of HC and OP patients are of paramount interest, and further in vivo validation study is needed. There is the possibility that OP patients could have their exfoliating deciduous teeth banked for future use in regenerative dentistry.

## 1. Introduction

Osteopetrosis (OP), also known as marble bone disease, comprises a group of rare heterogeneous hereditary disorders characterized by defective osteoclast activity where bones become sclerotic, thick, weak, and brittle. The condition was first described by the German radiologist Albers-Schonberg in 1904, and classically presents with increased bone density on radiographs [1].

There are several subtypes of OP that are classified according to their severity. The more severe forms, which occur in infancy, tend to be autosomal recessive inheritance (ARO) conditions, while the mildest forms are observed in adults and are inherited in an autosomal dominant manner (ADO). The overall incidence of these conditions varies, as the incidence of autosomal recessive osteopetrosis (ARO) is 1 in 200,000 births, while autosomal dominant OP (ADO) has an incidence of 1 in 20,000 births [2]. Patients with neonatal-onset often present with life-threatening complications such as bone marrow failure, while others might present with incidental findings of OP on radiographs. Classical ARO is characterized by repeated fractures, short stature, compressive neuropathies, hypocalcemia with attendant tetanic seizures, and life-threatening pancytopenia [3]. OP presents with dental abnormalities, including constriction of the canals that house neurovascular bundles along with obliteration of the dental pulp chambers (pulp stones) and bone marrow. These features usually contribute to bone necrosis and, hence, a high risk of osteomyelitis [4]

Of particular interest in this study is a specific intermediate recessive osteopetrosis (IRO) type which comprises a group of OP patients who present with renal tubular acidosis or carbonic anhydrase type II (CAII) deficiency, constituting a rare autosomal recessive disorder manifested clinically by renal tubular acidosis (RTA) and cerebral calcification. Other features include an increased frequency of fractures, short stature, dental abnormalities, cranial nerve compressions, and developmental delay [3,5,6,7,8]. The distinction between the variants of IRO can be trivial, as some patients have mutations in the *CA II* gene However, patients with *CA II* gene mutations have clinical features of brain calcification and renal tubular acidosis—hence the name “marble brain” syndrome. There are also some cases of IRO patients that harbor no known mutations but have all of the clinical features [9].

Human teeth harbor mesenchymal stem cells (MSCs) in the dental pulp, which contributes to tooth growth and repair [10]. The dental pulp of primary deciduous exfoliating teeth has been shown to be a significant progenitor/stem cell source (termed SHED, which refers to stem cells from human exfoliated deciduous teeth) due to the greater heterogeneous population of progenitor/stem cells residing in the pulp. Characterization of the cells has shown that SHED present positive expression for a set of embryonic stem cell markers (OCT4 and NANOG), stage-specific embryonic antigens (SSEA-3 and SSEA-4), mesenchymal stem cell markers (STRO-1 and CD146), and tumor recognition antigens (TRA-1-60 and TRA-1-81), but it is negative for the expression of hematopoietic markers such as CD45, CD11b/c, and HLADR [11]. Under specific culture conditions, these pluripotent-like cells differentiate into osteogenic, adipogenic, and neurogenic cells and osteodentin/bone-like structures [12,13,14]. Interest in stem cell therapies and regenerative medicines has grown notably over the last decade as their tremendous potential for effective treatments in a variety of medical diseases is being realized. Studies have shown that SHED are an ideal non-controversial and easily accessible cell source for use in regenerative medicine [15,16,17,18] The tissue regenerative effect of mesenchymal stem/stromal cells can be accomplished by the replacement of tissues with defects by differentiated dental MSCs/SCs themselves and/or through the induction of the de novo regeneration capacity of endogenous stem cells via remodeling of the stem cell niche [19].

It has been demonstrated that the subcutaneous transplantation of SHED MSSCs in nude mice can enhance bone repair via osteoclast inhibition in vivo [20]. SHED cells can also differentiate into vascular endothelial cells that form functional blood vessels by upregulating endogenous MEK1/ERK signaling [21].

SHED could have a potential application in periodontal tissue regeneration. Composites of allogeneic stem cell sheets derived from minipig deciduous teeth (SPD) combined with hydroxyapatite/tricalcium phosphate (HA/TCP) were transplanted into a periodontitis model in miniature swine to evaluate their therapeutic potential. The bone, cementum, and periodontal ligaments were regenerated in the periodontal defect area, and 75% of the samples underwent successful furcation regeneration in the SHED group [22].

Minimal risk of oncogenesis, high proliferative capacity, high multipotency, and immunosuppressive ability were exhibited [14,18].

SHED provides an unconventional and non-controversial source of human tissue that can be used for autologous stem cell therapy in a variety of medical diseases such as diabetes, spinal cord injury, and Parkinson’s disease [23,24]). Yang et al. suggested that there is a need for studies on the tissue-specific regeneration of oral and dental tissues in which the unique characteristics and functions of dental MSCs should be precisely demonstrated based on a comparison of dental MSCs with stem cells from other tissues [25].

Furthermore, SHED can easily be reprogrammed into induced pluripotent stem cells, representing an attractive, novel model for the investigation of pediatric diseases and disorders [26,27,28]. There is also a great need to grow MSCs from patients to overcome the stem cell transplant rejection barrier encountered with allotransplantation between different individuals [17].

Although there have been reports on SHED and their differentiation capabilities, there are very few studies in the literature that address whether human dental pulp progenitor/stem cells collected from subjects with a disease have the same multilineage capability for differentiation as their counterparts from healthy controls (HC). To our best knowledge, there are no reports on the properties of SHED from patients with OP or on whether osteopetrotic pulp may have a robust osteogenic capacity. Having such a property would make SHED attractive candidates for bone regeneration.

Proteomic analysis has the potential to identify novel biomarkers that complement currently existing protein stem cell surface markers. The discovery and development of a quantitative protein signature for dental pulp could simplify studies of mesenchymal stem/progenitor cells complicated by their inherent heterogeneity. Currently, definitive diagnosis of OP is mostly based on genetics and radiology tests, whereas definitive diagnostic protein biomarkers for OP have not yet been identified.

This study aimed to ascertain whether different mesenchymal stem/progenitor cells from human exfoliated deciduous teeth (SHED) obtained from OP and HC exhibit a similar multilineage capability in differentiation. In addition, we aimed to identify novel protein biomarkers for mesenchymal stem cells derived from the DP of HC and OP subjects as disease-specific or disease-related proteins.

## 2. Results

The workflow and results of this study are summarized in Figure 1. Briefly, mesenchymal cells were isolated from the dental pulp of deciduous teeth recently extracted from OP and HC subjects. The isolated cells were subsequently differentiated into osteocytes and chondrocytes. The cells were further characterized by quantitative proteomics using label-free quantitative liquid chromatography tandem mass spectrometry (LC–MS/MS).

### 2.1. Cell Proliferation and Colony-Forming Capacity of Mesenchymal Cells from Dental Pulp

The MTT assay and BrdU assay are two of the commonly used assay formats for the analysis of cell proliferation and viability. They are mostly single endpoint assays and are labor-intensive, requiring labeling lysis or fixation steps—characteristics which represent their main limitations. The xCELLigence real-time cell analysis is an excellent alternative system that provides quantitative data about cellular parameters, including the cell number, cell proliferation rate, cell size and shape, and the degree of cell adhesion as an index. It is a comparative dynamic analysis that ensures reproducible, real-time, label-free, and continuous readouts of cell proliferation, migration, and invasion of a given assay or between different assays of multiple cell types simultaneously. Proliferation was assessed using the Real-Time Cell Analyzer Dual Plate (RTCA-DP), a microelectronic biosensor system. All cell culture experiments were done on individual OP and HC samples representing biological replicates for each sample cohort and at least duplicate runs for each sample representing analytical replicates. The average value was calculated for each of the two sample groups, and we observed no statistically significant difference (*p* > 0.05) between the proliferation of cells from OP and HC subjects (Figure 2).

We only analyzed OP and HC dental pulp stromal cells for the colony assay as a complement to the cell proliferation assay. Equal cell numbers were seeded at a clonogenic density of 50 cells/cm^2^ (500 cells per well of a 6-well plate), and cells were cultured in α-minimal essential medium (α-MEM) supplemented with 20% FBS and 1% penicillin and streptomycin at 37 °C in a humidified incubator with 5% CO_2_. The medium was changed every 2–3 days. After one week in culture, the colonies were rinsed with PBS, fixed with 2% paraformaldehyde, and stained with 0.05% (*v*/*v*) crystal violet dye. The similarities in colony number and size between the two sample types were evaluated by visual observation and a quantitated plate scanner. There was no significant difference in the colony-forming ability of DP-MSCs derived from OP versus HC subjects. Representative images are shown in Figure 3.

### 2.2. Characterization of MSC-Like Cells from Dental Pulp

Characterization of MSSCs was done based on a panel of cell surface markers that was used and proposed by the International Society of Cellular Therapy (ISCT) for the minimal identification of human MSCs derived from bone marrow (BM) [29]. These immunophenotypes have also been adopted by other investigators to study MSCs in different tissues. We evaluated a total of 15 DP-MSSCs clone extracted from different human exfoliated deciduous teeth): 7 clones from 3 osteopetrosis patients and 8 clones from 3 normal healthy donors. These were propagated in our laboratory, as depicted in Figure 4. All clones were at passage 3. The 15 clones showed similar characteristics, and all of the DP-MSSCs were found to be negative for the hematopoietic markers CD14, CD34, CD45, MHC-II, and CD133, as shown by our negative cocktail. Additionally, all of the MSSC cells were found to be strongly positive for CD 44, CD73, CD90, and CD105 (Figure 4).

### 2.3. Cell Differentiation of Dental Pulp Mesenchymal Cells

Isolated cells from OP and HC were differentiated for 21 days into osteocytes and chondrocytes, as confirmed by alkaline phosphatase (ALP) and Alcian blue staining, respectively.

Cells derived from OP and HC were successfully differentiated into osteocyte and chondrocyte lineages (Figure 5).

### 2.4. Protein Expression Signatures of Undifferentiated DP MSCs

Whole-cell lysates of bulk/unsorted and undifferentiated mesenchymal stem/stromal cells from OP and HC subjects were subjected to label-free quantitative proteomics analysis. All samples from patients were gender and age-matched with samples from HC subjects (Table 1). This analysis was taken as a baseline protein profile to examine the similarities or differences between DP-MSSCs derived from OP and HC subjects. We analyzed the total protein content of the bulk/unsorted and undifferentiated DP mesenchymal cells from OP and HC subjects without any form of differentiation. The rationale was to determine the protein content prior to differentiation. This allowed us to see what proteins were being influenced by cell differentiation.

A total of 1499 unique proteins from undifferentiated whole cell lysates of HC and OP subjects were identified. Only 86 proteins (5.7%) differed significantly in abundance between HC and OP samples (*p* < 0.001 and >2-fold change, implying a 94.3% similarity in global protein fingerprints between the two sample groups (Appendix A). The changes in abundance of this protein dataset reflect the degree of similarity in changes to polypeptide production between the two sample groups.

### 2.5. Protein Signatures Related to Osteogenesis in HC and OP DP-MSSCs

MSSCs from passage 3, were extracted from OP and HC and were differentiated into chondrocytes and osteocytes. The cells were subjected to in-solution tryptic digestion and characterized by quantitative proteomic analysis. Similarly to undifferentiated cell protein profiles, we observed a remarkable level of similarity in the protein profiles between differentiated osteogenic cells from OP and HC subjects. Compared with undifferentiated cells, a total of 1399 unique protein species were identified between undifferentiated/osteogenic/chondrogenic HC and OP, of which the levels of 303 proteins (21.7%) differed significantly between all six cell types of HC samples (*p* < 0.001 and >2-fold change (i.e., undifferentiated/osteogenic/chondrogenic cells from HC and undifferentiated/osteogenic/chondrogenic cells from OP). The profiles of a total of 79 proteins (5.6%) differed significantly among undifferentiated, osteogenic, and chondrogenic differentiated HC cells (Appendix A). A similar analysis was conducted for OP samples consisting of undifferentiated, osteogenic, and chondrogenic differentiated cells, and a greater fraction of 196 proteins (14%) showed marked changes in abundance (Appendix A). It is important to note that the DP-MSSCs were separately differentiated into chondrocytes and osteocytes. The individual resulting differentiated cells as well as the bulk/undifferentiated cells were individually lysed and analyzed to determine their changes in protein abundance, as reported above.

### 2.6. Cellular and Functional Annotations of the Identified DP-MSSCs

We further explored the cellular and functional annotations of some of the identified proteins using Ingenuity Pathway Analysis (IPA, Qiagen, MD, USA). The 86 proteins in the dataset that were significantly differentially expressed between the bulk undifferentiated OP and HC samples were evaluated to determine their involvement in different signaling networks using the IPA database. Interestingly, some of the molecules were implicated in the signaling network with cellular involvement in pathways including MAPK, ERK 1/2, PI3K, and integrin, among others, as indicated in Figure 6A and Table 2.

A similar analysis was done for 196 proteins with marked expression changes between undifferentiated, osteogenic, and chondrogenic differentiated MSSCs in both OP and HC cells. Many of these proteins were implicated in different signaling networks involving bone and connective tissue disorders (Figure 6B). Surprisingly, the observed changes in both datasets were OP-disease-related rather than being associated with the stem cell signaling network. This further indicates that DP-MSSCs derived from both OP and HC share highly similar properties.

## 3. Discussion

There is a paucity of literature that addresses whether human dental pulp MSSCs (DP-MSSCs) obtained from subjects with disease have the same multilineage capability in differentiation as human DP-MSSCs from healthy controls (HC). To our best knowledge, this is the first comprehensive comparative proteomic analysis that has attempted to determine whether MSCs from human dental pulp obtained from patients diagnosed with osteopetrosis (OP) have the same multilineage differentiation capability as DP-MSSCs from HC.

We characterized DP-mesenchymal stem/progenitor cells from human exfoliated deciduous teeth (SHED) and the commercially available SHED cell line Axol ax3901 cells using multiple analysis platforms, including flow cytometry and proteomics by label-free quantitative liquid chromatography tandem mass spectrometry (LC–MS/MS).

Characterization of MSSCs was conducted based on a panel of cell surface markers that were proposed by the International Society of Cellular Therapy (ISCT) for the identification of human MSSCs derived from bone marrow (BM). It is well-known that DP cells, upon culture, yield a population of cells with characteristics typical of MSCs, including clonogenicity and expression of essential MSC surface markers [10,11,30].

Immunophenotypic analyses by flow cytometry gave very similar results with strong expression of some commonly used BM-derived MSCs markers, including CD44, CD73, and CD90 and negative results for CD34 and CD45 hematopoietic stem cell markers. Furthermore, MSSCs from both OP and HC were successfully induced to undergo osteogenic and chondrogenic differentiation (Figure 3).

Although the present study was based on cells isolated from osteopetrosis patients and healthy subjects, our findings are similar to those obtained in a recent study where iPSCs derived from OP patient BM were found to express pluripotent stem cell markers and demonstrate trilineage differentiation potential [31].

Our analyses revealed that DP-MSSCs from patients and HC subjects shared a large number of features (>95%) in their differentiation and protein expression profiles. In a similar study, iPSCs were derived from urine samples of autosomal dominant OP patients (ADO2-iPSCs) and normal control iPSCs (NC-iPSCs) and subjected to proteomic analysis. The majority (97.3%) of the 6359 identified proteins showed similar expression profiles between the two different cell lines [20]. In this present study, about 1400 proteins were identified, of which only 5.7% had marked expression changes from the two analyzed undifferentiated DP-MSSCs from HC and OP patients. Very similar to undifferentiated cells, only 5.6% of the identified proteins were found to differ significantly between OP-DP-MSC undifferentiated/osteogenic/differentiated cells. This finding suggests that HC- and OP-derived SHED cells are very similar. We acknowledge that the protein expression changes in the above cited paper might not be the same as the changes observed in this presented study. As we are mindful of this, we focus on the similarities in the fractions of differentially expressed proteins as well as the homogeneity in the fractions of housekeeping proteins in our study and the referred study, rather than comparing the lists of observed changes between the two studies.

Currently, stem cell therapy is providing hope for a cure for many diseases that were previously considered incurable. Several studies have demonstrated the great differentiation potential of dental pulp MSSCs and their ability to remain active throughout life and proliferation of odontoblasts in dental repair [18]. These features make them potentially useful for cell therapy for a variety of diseases, including diabetes, arthritis, neurodegeneration, brain, spinal cord injuries, and cancer. MSCs derived from bone marrow transplants are widely used for a number of malignant and non-malignant hematological disorders. However, issues associated with donor/recipient human leukocyte antigen HLA incompatibility often preclude their usefulness. DPSSCs can be potentially and reproducibly expanded in vitro within a short period of time with the potential to effectively reconstitute defective dental tissue for therapeutic use [32,33,34]. The advent of DP-MSSCs from OP with similar differentiation properties as DP-MSSCs from HC has introduced the possibility of preserving their exfoliated deciduous teeth for future use in autologous cell therapy as opposed to allogenic therapy, where the donor and recipient of the stem cells are different (allogeneic grafts), a process that can lead to rejection due to HLA mismatch. Therefore, our finding of highly similar (94.4%) protein expression signatures in the DP-MSSCs of OP and HC subjects gives hope to the idea of prospective banking of DP-MSSCs from exfoliated deciduous teeth of OP patients for future MSC therapy and/or potential tissue re-engineering. 

Despite this remarkable similarity, we investigated the subtle differences in protein expression to determine whether they are related to stemness or the primary OP disease.

Some of the identified differentially expressed proteins were implicated in signaling networks involving bone and connective tissue disorders as mapped in the database of the Ingenuity Pathway Analysis program (IPA; Figure 6A). The core of the network shown in Figure 6A includes MAPK, ERK 1/2, PI3K, and integrin, among others. Interestingly, similar to what has been previously described, activation of ERK1/2 and phosphatidylinositol phosphate (PI3K) signaling enhances osteoblastogenesis and osteoblast survival [35,36]. 

Among the proteins identified in the network in this study are those that are indirectly related to TGF-β, EGF, and ERK1/2. These proteins are known for their roles as regulators of cellular proliferation, migration, and differentiation of osteoblasts, chondrocytes, and osteoclasts [35,37,38,39].

Developed bone is made up of an organic matrix (osteoid) consisting mainly of collagen produced by osteoblasts [35]. We observed a 2.23-fold greater change in the level of expression of collagen type 1A1 in HC samples compared with OP samples.

FERMT2 (fermitin family homolog 2) is a scaffolding protein that enhances the activation of integrin and the regulation of bone homeostasis [15]. Integrin has been associated with the activation of ERK, a process critical for the induction of bone-formation-related genes in osteoblast-like cells [40].

A more recent study demonstrated that a decrease of more than 2-fold in the expression of kindlin-2 impairs osteoblast formation and function in mice [26]. We found that a reduction of more than 6-fold in the expression of FERMT2 in OP compared with HC is associated with a lack of activation of osteoclasts for proper bone formation in the OP.

Studies have showed that overexpression of GNA13 is associated with increased proliferation and tumorigenicity of gastric cancer cells through the upregulation of c-Myc, activation of AKT, and activity of the ERK signaling pathway [41,42,43]. OP had 2.5-fold greater GNA13 expression than HC subjects. Figure 6A shows that GNA13 expression is directly connected with G protein alpha and indirectly connected to integrin and ERK1/2; thus, its underexpression in OP patients would lead to impaired activation of integrin and ERK1/2.

Type III collagen is a fibrillary collagen that is thought to be secreted by fibroblasts and other mesenchymal cell types. Col3 has also been demonstrated to play a key role in the development of trabecular bone by exerting effects on osteoblast differentiation [44]. In this study, we observed a more than 6-fold reduction in the expression of COL3A1 in OP compared with HC. It is not surprising that the majority of our studied OP patients presented with craniofacial and skeletal phenotypic features considering the important role of COL3 in osteoblastogenesis [44].

## 4. Materials and Methods

### 4.1. Patient Materials

The mesenchymal stromal cells were derived from the pulp of primary incisor and canine teeth from HC and OP subjects, aged 6–15 years—the normal exfoliation age period. Patients seeking care at the dental clinic of the King Faisal Specialist Hospital and Research Center were recruited. Inclusion criteria were diagnosis of autosomal recessive OP with renal tubular acidosis due to carbonic anhydrase II deficiency. Similarly, HC subjects were recruited when they sought dental evaluation and treatment at the hospital. Subjects who had diseased teeth with caries or abscesses were excluded from the study. All patients and subjects completed a written and signed consent form. The study was conducted in accordance with the Declaration of Helsinki, and the protocol was approved by the Office of Research Advisory Council (RAC)/Office of Research Affairs (ORA), King Faisal Specialist and Research Center (12-BIO 2343-20), May 2016.

A total of 24 deciduous teeth consisting of central, lateral incisors, and canines were extracted from 5 OP and 6 HC subjects. The clinical characteristics of all patients and control subjects are summarized in Table 1.

### 4.2. Cell Isolation, Culture, and Differentiation of Dental Pulp Mesenchymal Cells

Mesenchymal stromal cells were isolated from the dental pulp of recently extracted deciduous teeth from HC and OP subjects, as previously described [12]. Briefly, exfoliated teeth were extracted as part of the treatment plan, cleaned with 5 cc chlorhexidine gluconate 0.12%, and immediately transported to the research laboratory on ice at 4 °C. Teeth were placed in wash buffer (PBS, 1% gentamicin and 1% fungizone) on ice in a container. Each tooth was kept in solution for 1–2 min and washed three times. Each tooth was carefully opened from the bottom using a low dental handpiece to gently expose the pulp. The attached tissue was removed using a sterile blade under the cell culture hood. Using a barbed broach, as previously described [14], the extracted dental pulp was gently diced into tiny pieces before enzymatic digestion. The tissue was then treated with digestive medium containing 3 mg/mL type I collagenase (Gibco, MA, USA), 4 mg/mL dispase (Gibco, MA, USA), and 1% collagenase/hyaluronidase (Stem Cell Technologies, Vancouver, BC, Canada) for 1–1.5 h at 37 °C. The digestion process was terminated by MSC-FBS, and the sample was centrifuged at 1500 rpm for 5 min (4 °C). The pellet was resuspended in fresh growth medium (α-MEM (no phenol) (Gibco, MA, USA) with 20% MSC-FBS (Gibco, MA, USA), 1% non-essential amino acids (Sigma), 1% AA (Sigma, St. Louis, MO, USA), 1% ITS-A (Gibco), 2 μM L-glutamine (Sigma, St. Louis, MO, USA), 100 μM L-ascorbic acid 2-phosphate (Stem Cell Technologies, Vancouver, BC, Canada), 100 ng/mL EGF (Gibco, MA, USA), and 40 ng/mL FGF(Gibco, MA, USA). The sample was then placed in a 12-well petri dish pre-coated with 10 µg fibronectin (Corning) for at least 2 h at 37 °C, supplemented with culture medium, and placed in an incubator at 37 °C with 5% CO_2_.

The medium was changed after 2–3 days for 5–7 days until 80%–90% confluence was reached, and cells were collected and frozen using MSC freezing medium (Stem Cell Technologies, Vancouver, BC, Canada) and then transferred to liquid nitrogen storage until being required for further analysis.

### 4.3. Cell Proliferation Assay

MTT and BrdU are two of the most commonly used assays for the analysis of cell proliferation and viability. They are mostly single endpoint assays and are labor-intensive, requiring labeling lysis or fixation steps, factors which constitute their main limitations. xCELLigence real-time cell analysis is an excellent alternative system that provides quantitative data about cellular parameters including the cell number, cell proliferation rate, cell size and shape, and cell adhesion index. It provides a comparative dynamic analysis ensuring reproducible, real-time, label-free, and continuous readouts of cell proliferation, migration, and invasion of the same assay or between different assays of multiple cell types simultaneously. We used an inverted microscope and the microelectronic biosensor system Real-Time Cell Analyzer Dual Plate (RTCA-DP) for the cell-based assays. Equal numbers of OP and HC cells per well were seeded into 100 µL of media in 96-well microplates (E-Plate 96), and the proliferation of the cells were monitored and measured (cell index) over time using the xCELLigence system. All cell culture experiments were done on individual OP and HC samples representing biological replicates for each sample cohort and at least duplicate runs for each sample representing analytical replicates. Similarities or differences in cell viability, cell number, cell morphology, and degree of adhesion were evaluated, and growth conditions were monitored for 4 days.

### 4.4. Dental Pulp Mesenchymal Stem Cell Differentiation

The quality and reproducibility of different passages of the cultured dental pulp mesenchymal cells were assessed by 2D gel electrophoresis. This revealed that passage 3 and 4 were highly similar in terms of both the total number of protein spots and the resolution of 2-DE gel images, as previously described [45]. Therefore, all differentiations, flow cytometry, and proteomic analyses were done using cells in passage 3.

Cells from OP and HC were subjected to differentiation into chondrocytes and osteocytes, as previously described [46,47,48], using appropriate differentiation media.

For osteogenic differentiation, cells were plated at a concentration of 1 × 10^5^ cells per well in the fibronectin-coated tissue culture plate and subsequently induced with MSC osteogenic differentiation medium (Lonza, Basel, Switzerland) using MSC growth medium. They were incubated for 12–14 days, and the medium was changed every 2–3 days. Formation of bone mass was detected using the bright orange–red dye Alizarin Red S.

For chondrogenic differentiation, Axol ax3901 dental pulp stem cells (available at https://www.axolbio.com) were re-suspended in dental pulp stem cell culture medium pre-warmed to 37 °C at a concentration of 1.6 × 10^7^ cells/mL. Cells were seeded as 5 μL drops of cells into multiwell plates incubated for 2 h at 37 °C in a humidified incubator with 5% CO_2_. MSC chondrogenesis medium (Lonza, Basel, Switzerland) pre-warmed at 37 °C was gently added to the wells (2 mL per well for a 6-well plate). The culture medium was changed every 2–3 days until day 21. Cells were fixed with 4% paraformaldehyde and stained for sulfated proteoglycans using Alcian blue stain.

### 4.5. Flow Cytometry

Dental pulp mesenchymal stem cells (DPMSCs) were phenotypically characterized using flow cytometry, as previously described [29]. Briefly, 1–5 × 10^5^ harvested cells were stained with monoclonal antibodies designated as required for the minimal identification of human MSCs derived from bone marrow, as proposed by the International Society of Cellular Therapy (ISCT) and adopted by others as well. These antibodies are CD44, CD73, CD90, and CD105 for mesenchymal stem cells and CD14, CD34, CD45, MHC-II, and CD133 for the exclusion of hematopoietic cells. The expression of the corresponding cell surface proteins was assayed by flow cytometry using the BD FACS Aria (Becton Dickinson, NJ, USA). The list of antibodies used in the study (clones, colors, and vendor) is summarized in Appendix A with a description of the Human MSC analysis Kit by BD StemflowTM/BD Biosciences.

### 4.6. Cell Differentiation, Expression Proteomics, and Biomarker Discovery for SHED

We conducted an overview and schematic illustration of our multiplatform analysis strategies as an efficient comparative analysis and biomarker discovery tool for dental pulp mesenchymal stem cells from OP and HC subjects. Proteins were characterized by quantitative mass spectrometry (LC–MS/MS) and functional annotation of identified significantly differentially expressed proteins by signaling pathway analysis.

### 4.7. Colony Forming Potential of Isolated Dental Pulp Mesenchymal Cells

The colony forming ability of DP-MSSCs derived from OP and HC subjects was evaluated as previously described [12,13,49]. Briefly, equal numbers of cells from OP and HC subjects were seeded into six-well plates at a density of 500 cells/well and incubated at 37 °C for 10 days in α-minimal essential medium (α-MEM) supplemented with 20% FBS and 1% antibiotics at 37 °C in a humidified incubator with 5% CO_2_.

The medium was changed every 2–3 days for about a week. The medium was discarded, and 500 µL of 3.7% formaldehyde (in 1 × PBS) was added to each well, and this was incubated for 15 min at room temperature. Cells were washed once with 1 mL 1× PBS and subsequently stained with 0.1% crystal violet (Santa Cruz, TX, USA).

### 4.8. Protein in-Solution Digestion

All samples were individually accessed using 2-DE gels as a form of quality control in the selection of samples within a cohort group (data not shown) accounting for biological replicates. Thereafter, due to the inherent limitation of low throughput of MS-based analysis, representative samples within a cohort group were pooled and subjected to MS analysis, and each pooled sample cohort was run in triplicate for analytical replicates, as previously described [45]. For each isolated mesenchymal cell from the dental pulp of recently extracted deciduous teeth, an equal amount of 100 μg complex protein mixture was taken from HC and OP pooled samples and subjected to in-solution protein digestion prior to LC–MS/MS analysis, as previously described [45,50]. Briefly, proteins were first denatured in 0.1% RapiGest SF (Waters Corp., Manchester, UK) at 80 °C for 15 min, reduced in 10 mM DTT at 60 °C for 30 min, and alkylated in 10 mM iodoacetamide (IAA) for 40 min at room temperature in the dark. Samples were trypsin digested at 37 °C overnight (50:1, sample/trypsin ratio) (Promega, WI, USA). Samples were diluted with aqueous 0.1% formic acid prior to LC–MS/MS analysis in order to achieve a concentration of approximately 1 μg/μL. All samples were spiked with yeast alcohol dehydrogenase (ADH; P00330) as an internal standard for absolute quantitation. 

### 4.9. Protein Identification by Liquid Chromatography Mass Spectrometry (LC–MS^E^) Analysis

The digested peptides from different cell lysate samples were subjected to 1D Nano Acquity liquid chromatography coupled with tandem mass spectrometry on Synapt G2 (Waters Corp, Manchester, UK), as previously described [45]. A total of 3 μL of sample, representing ~3 μg of protein digest, was injected onto a Trizaic column, and samples were infused using the Acquity sample manager with a mobile phase consisting of A1 99% water +1% acetonitrile + 0.1% formic acid and B1 acetonitrile + 0.1% formic acid with a sample flow rate of 0.450 μL/min. Data acquisition was performed using iron mobility separation experiments (HDMS^E^), and data were acquired over a range of *m/z* 50–1300 Da with a total acquisition time of 115 min. All samples were analyzed in triplicate runs (triplicate runs were repeated on different occasions as a measure of reproducibility), and data were acquired using the MassLynx program (version. 4.1, SCN833; Waters, Manchester, UK) operated in resolution and positive polarity modes. Progenesis QI for proteomics (Progenesis QIfp version 3.0) (Nonlinear Dynamics/Waters, Manchester, UK) was used for all automated data processing and database searching. The generated peptide masses were searched against two unified non-redundant databases (Uniprot/SwissProt Human protein sequence database) using the Progenesis QIfp for protein identification (Waters).

### 4.10. Data Analysis and Informatics

All generated quantitative proteomic expression analysis was processed by Progenesis QI v.3.0.6039 for proteomics. The species-specific human protein database from Uniprot of reviewed non-redundant data containing thousands of entries was used to process and search the data to accurately quantify and identify proteins that were differentially expressed significantly between sample groups. The generated differentially expressed data were filtered to show only statistically significantly differentially expressed proteins (ANOVA, *p* ≤ 0.05) and a fold change ≥ 2.0. In order to achieve absolute quantification of identified protein changes, the quantitation of ADH as an internal standard was applied, as incorporated in the Progenesis QIfp (Nonlinear Dynamics/Waters, Manchester, UK).

## 5. Conclusions

Several studies have demonstrated the high differentiation potential of MSCs, thus making them attractive targets for cell therapy for a variety of diseases. Bone marrow transplant derived MSCs are widely used for the treatment of a number of blood disorders, including cancer. However, issues associated with donor/recipient HLA incompatibility often preclude their usefulness. Our finding that DP-MSSCs from OP have similar differentiation properties to DP-MSSCs from HC might indicates that preserving exfoliating deciduous teeth for future use in autologous cell therapy may be worthwhile once the findings are further validated. This presents a significant advantage compared with allogenic therapy, where the donor and the recipient of the stem cells are different individuals, increasing the risk of rejection due to incompatibility.

Furthermore, we analyzed both qualitative and quantitative immunophenotypic and proteomic expression data to assess the homogeneity in DP-MSCs. Our results show similar patterns of proliferation and differentiation as well as protein expression between DP-MSCs obtained from OP and HC subjects. The average percentage correlation from all pairs of OP and HC subject samples was 0.944, indicating a high degree of similarity between them.

Our finding that DP-MSSCs from HC and OP subjects have similar protein expression signatures is of paramount interest, and future validation in vivo may lead to OP patients being offered the possibility to have their exfoliating deciduous teeth banked (rather than discarded, as is the current practice) for future autologous stem cell therapy instead of soliciting for a matched donor for allogeneic stem cell therapy, which has associated risks of GvHD and eventual failed engraftment.

## Figures and Tables

**Figure 1 ijms-22-00380-f001:**
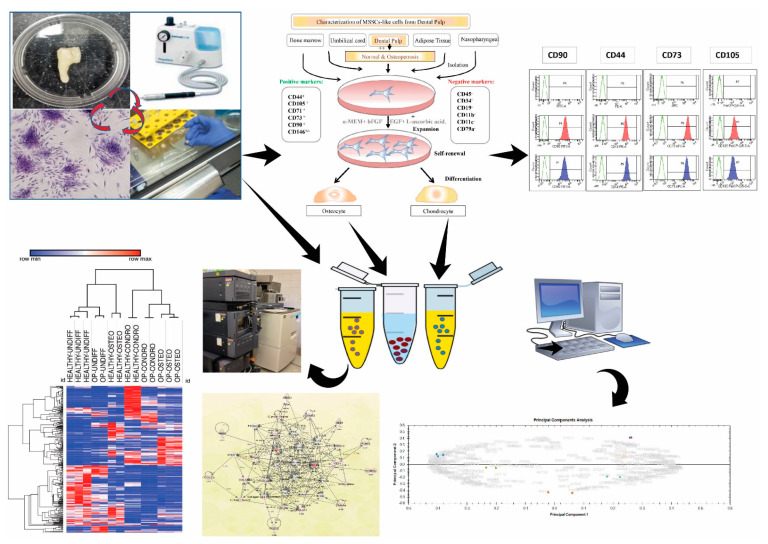
Workflow illustration: Mesenchymal cells were isolated from the dental pulp of recently extracted deciduous teeth from osteopetrosis (OP) patients and healthy control (HC) subjects. The dental pulp mesenchymal cells were differentiated into osteocytes and chondrocytes. Cells were further characterized using by proteomics using label-free quantitative liquid chromatography tandem mass spectrometry (LC–MS/MS). The details of the clinical characteristics of all the samples are listed in Table 1.

**Figure 2 ijms-22-00380-f002:**
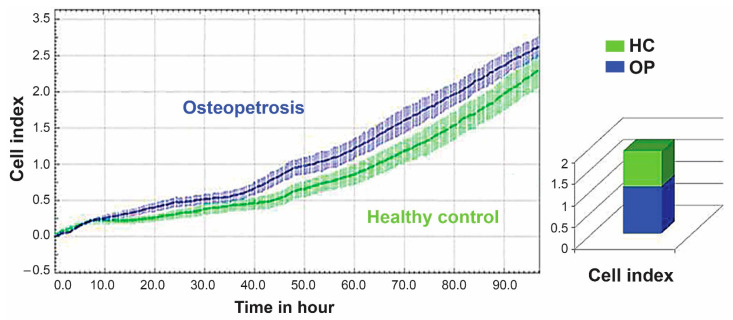
Growth curves showing similarities in proliferation of dental pulp mesenchymal stem cells (DP-MSSCs) from HC and OP using the microelectronic biosensor system Real-Time Cell Analyzer Dual Plate (RTCA-DP). Equal numbers of OP and HC cells per well were seeded into 100 µL of media in 96-well microplates (E-Plate), and proliferation of the cells was monitored and measured (cell index) over time using the xCELLigence system. The panel illustrates one of the two independent representative OP and HC cell cultures done at two different time points showing highly similar growth curves. The histogram shows the average quantitation of the cell index of each cell type. The images were partly generated by xCELLigence RTCA-DP.

**Figure 3 ijms-22-00380-f003:**
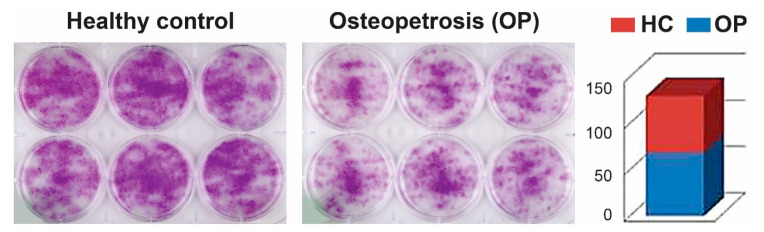
Similarity in the colony-forming potential of DP-MSCs from OP and HC. Equal cell numbers from OP and HC subjects were seeded at a density of approximately 500 cells/well and incubated at 37 °C for 10 days. Cells were washed and subsequently stained with 0.1% crystal violet (Santa Cruz) for visualization. Individual samples in each of the analyzed cohort were considered as biological replicates of that group, while at least duplicates runs of each of the pairs of OP and HC were considered as analytical replicates in order to demonstrate the reproducibility of the colony forming potentials of the two cell types from OP and HC. The histogram shows the quantitation of staining as visualized under the microscope. The average intensity of each of the 6 wells was used to generate the histogram. The variation was very minimal and had no statistically significant difference (*p* > 0.05), so we used stack plots rather than individual bars. The y-axis shows the quantitation as a measure of the optical density of the scanned wells, and the x-axis gives the number of replicated cell cultures.

**Figure 4 ijms-22-00380-f004:**
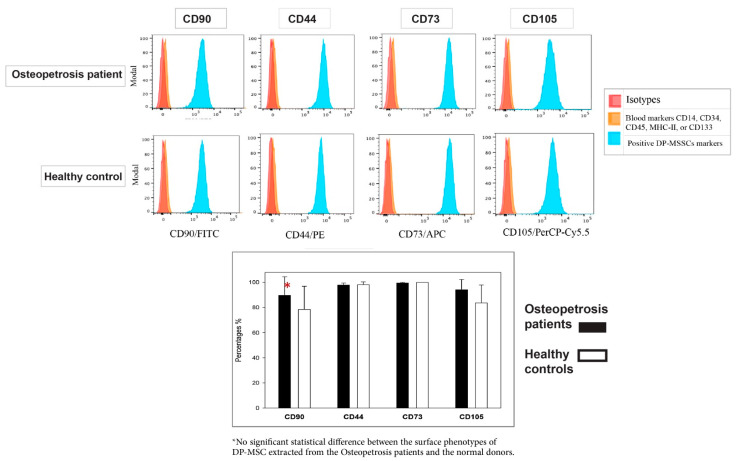
The comparisons of surface markers of DP-MSSCs from normal donors and osteopetrosis patients using flow cytometry. Representative histograms showing the expression of MSC surface markers (CD90, CD44, CD73, and CD105) as analyzed by flow cytometry. The graph shows a comparison of MSC markers (average mean ± 95% confidence intervals) of 15 clones of DP-MSCs extracted from different human exfoliated deciduous teeth. Seven clones from osteopetrosis patients (black bars) and eight clones from normal donors (white bars) were analyzed by flow cytometry. Means and percentages are given, and error bars are 95% confidence intervals. In the green histograms (isotype), samples from both normal and patients were negative for hematopoietic markers (CD14, CD34, CD45, MHC-II, or CD133) (blue histograms). The blue histograms show positive markers in isolates from normal donors (8) and osteopetrosis patients (7), and negative blood markers are represented by orange and pink histograms as isotope controls. * indicates no statistical significance (*p* > 0.05).

**Figure 5 ijms-22-00380-f005:**
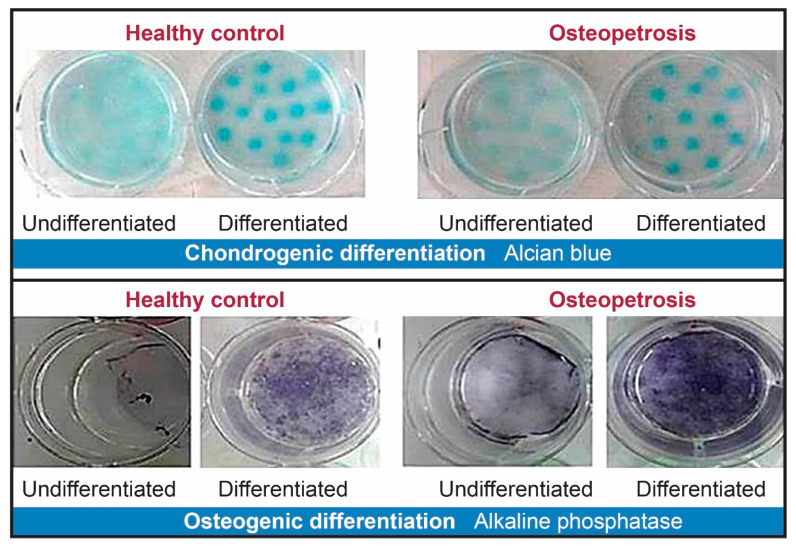
DP-MSC cell differentiation. Upper panel: Chondrogenic differentiation of unsorted bulk DP-MSSCs. Alcian blue staining of undifferentiated and osteogenic differentiated cells in HC and OP subjects after the induction of cells to undergo osteogenic lineage differentiation for 21 days. Lower panel: Osteogenic differentiation of bulk unsorted DP-MSSCs with alkaline phosphatase staining of undifferentiated and osteogenic differentiated cells from HC and OP subjects after induction of cells to undergo osteogenic lineage differentiation for 21 days.

**Figure 6 ijms-22-00380-f006:**
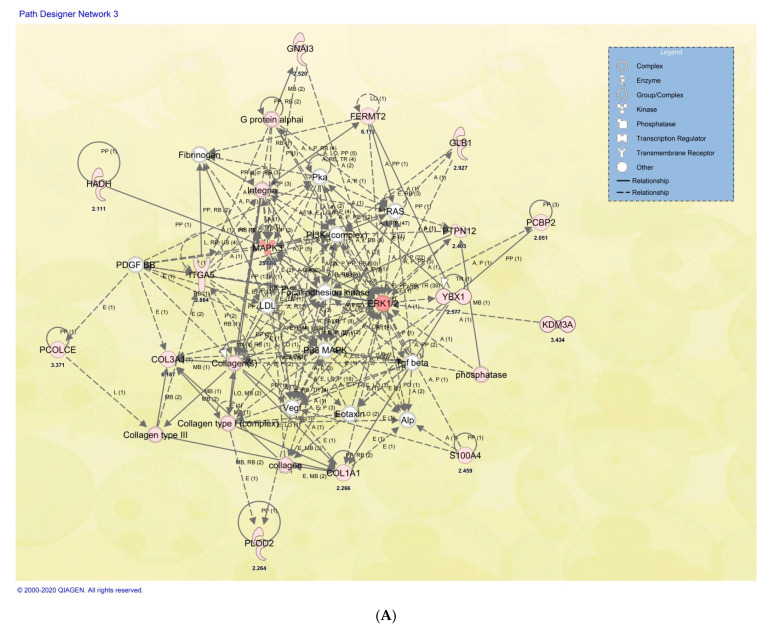
(**A**) Ingenuity Pathway Analysis (IPA) of the 86 proteins in the dataset that were found to be significantly differentially expressed between the bulk/undifferentiated HC and OP samples. Some of the identified proteins (highlighted in pink) were shown to have cellular involvements in signaling network pathway interactions. In the core of the network (in red), interactions of MAPK, ERK 1/2, PI3K, and integrin connected to osteopathogenesis were identified. Proteins in grey and white colors represent molecules from the IPA database that make up the network. (**B**) Ingenuity Pathway Analysis (IPA) of 196 proteins with marked expression changes between undifferentiated, osteogenic, and chondrogenic differentiated MSSCs in both OP and HC cells. Some of these proteins (highlighted in pink, as described above) were implicated in different signaling networks involving bone and connective tissue disorders. The network analysis was generated through the use of IPA (QIAGEN Inc., https://www.qiagenbioinformatics.com.

**Table 1 ijms-22-00380-t001:** Clinical characteristics of study patients.

Patient	Age (Years)	Gender	Mode/Gene	Developmental	Cranial/Bone	Renal
DP#42-5242	9	M	CAII (IVS2 + 1G > A)	Delayed	Fracture ++	RTA
DP#47-6342	8	F	AR-CAII c.232 + 1G > A, IVS2 + 1G > A	Short Stature	Calcification	RTA
DP#52-4774	14	F	CAII	ADHD, Delayed	Calcification	RTA
DP#53-3492	13	M	CAII	Delayed	Calcification, Fractures	RTA
DP#54-5544	7	M	AD-CAII c.232 + 1G > A; IVS2 + 1G > A	Delayed	Calcification, Fractures	RTA
DP#31-7530	9	F	Healthy	Normal	No Abnormalities	N/A
DP#41-1082	7	F	Healthy	Normal	No Abnormalities	N/A
DP#46-7880	8	M	Healthy	Normal	No Abnormalities	N/A
DP#50-4300	14	M	Healthy	Normal	No Abnormalities	N/A
DP#51-5231	7	M	Healthy	Normal	No Abnormalities	N/A
DP#57-4600	11	F	Healthy	Normal	No Abnormalities	N/A

CA-II, carbonic anhydrase II; RTA, renal tubular acidosis; ADHD, attention deficit hyperactivity disorder; AR, autosomal recessive; AD, autosomal dominant; *IVS 2*, intron 2.

**Table 2 ijms-22-00380-t002:** Some of the 86 proteins in the dataset that were significantly differentially expressed between the bulk undifferentiated HC and OP samples were mapped in Ingenuity Pathway Analysis (IPA). Their functional annotations are indicated, as are their involvements in osteogenesis as well as in different bone and connective tissue disorders.

Symbol	Entrez Gene Name	GenPept/ UniProt/Swiss-Prot Accession	Expr Fold Change OP:HC	Location	Family	Bone Involvement
COL1A1	Collagen type I alpha 1 chain	P02452	−2.26587705	Extracellular Space	Other	Bone matrix/Osteoblast
COL3A1	Collagen type III alpha 1 chain	P02461	−6.1868361	Extracellular Space	Other	Bone matrix/Osteoblast
FERMT2	Fermitin family member 2	Q96AC1	−6.11159085	Cytoplasm	Other	Bone homeostasis regulation
GLB1	Galactosidase beta 1	P16278	2.92715725	Cytoplasm	Enzyme	
GNAI3	G protein subunit alpha i3	P08754	−2.52033996	Cytoplasm	Enzyme	Impaired activation of integrin and ERK1/2.
HADH	Hydroxyacyl-CoA dehydrogenase	Q16836	2.1108754	Cytoplasm	Enzyme	In OP gene panel
ITGA5	Integrin subunit alpha 5	P08648	2.88383695	Plasma Membrane	Transmembrane Receptor	osteoclastogenesis
KDM3A	Lysine demethylase 3A	Q9Y4C1	3.43441641	Nucleus	Transcription regulator	OP associated
MAPK3	Mitogen-activated protein kinase 3	P27361	25.6856293	Cytoplasm	Kinase	Bone Diseases
PCBP2	Poly(rC) binding protein 2	Q15366	2.05142965	Nucleus	Other	
PCOLCE	Procollagen C-endopeptidase enhancer	Q15113	3.37138722	Extracellular Space	Other	osteoclast–osteoblast bone modeling
PLOD2	Procollagen-lysine,2-oxoglutarate 5-dioxygenase 2	O00469	2.26356027	Cytoplasm	Enzyme	bone mass syndromes
PTPN12	Protein tyrosine phosphatase non-receptor type 12	Q05209	2.40278726	Cytoplasm	Phosphatase	Bone resorbing
S100A4	S100 calcium binding protein A4	P26447	2.45896457	Cytoplasm	Other	Bone resorption osteoclast function
YBX1	Y-box binding protein 1	P67809	2.57704761	Nucleus	Transcription regulator	

The fold change was calculated based on the expression values of OP/HC. Therefore, the positive fold-change values mean upregulation in OP compared with HC and vice versa for negative values, i.e., upregulated in HC compared with OP.

## Data Availability

All data generated or analyzed during this study are included in this published article and its Appendix A files. All methods are as detailed in this manuscript. The accession number for the protein identification and characterization data reported in this paper is based on Uniprot format. The raw mass spectrometry data generated in this study using Waters Synapt G2 HDMSE would be available from the corresponding author on reasonable request.

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
