# Peer review of "Proteomic Profiling of the First Human Dental Pulp Mesenchymal Stem/Stromal Cells from Carbonic Anhydrase II Deficiency Osteopetrosis Patients"

_ijms, 2020, doi:10.3390/ijms22010380_

Round 1

Reviewer 1 Report

The authors sought to characterize the role of mesenchymal stem /stromal cells in osteopetrosis patients. I felt that a lot of effort was contributed to achieve this goal, and this manuscript is clear logic and well-written. However, it is mandatory to explain/correct the manuscript in some points.  

  1. How many biological / analytical replicates were performed?
  2. Any correction on the p-value?

Reviewer 2 Report

This study verified if different types of SHED obtained from osteopetrosis and heathy control exhibit a similar differentiation potential. The authors also identified novel disease-specific protein biomarkers. The study design is appropriated and clinically related. Moreover, the manuscript is well written and the results are informative. Overall, it is a nice study. In this respect, I do believe that this revised manuscript is worthy of being published in the journal in the present form.

Thank you for the opportunity.

Author Response

Please see the attachment for reviewer #1 above.

Not applicable for this reviewer#2

This manuscript is a resubmission of an earlier submission. The following is a list of the peer review reports and author responses from that submission.

Round 1

Reviewer 1 Report

The paper by Alkhayal at al. focuses on interesting aspects of biological and functional differences between dental pulp derived mesenchymal stem/ stromal cells (DP-MSCs) harvested from healthy donors and patients suffering with osteopetrosis. Although the paper includes interesting and novel proteomic analyses of both DP-MSC fraction, there are several major concerns, mistakes and limitations visible in the manucript, which have to be considered and corrected prior to potential publication in IJMS.

Major comments:

  • There are several major mistakes in English writing. The language should be corrected in whole manuscript (professional English language editing services are recommended).
  • Line 19-20: The Authors should provide appropriate name of MSCs in whole manuscript. The authors use several names “mesenchymal cell”, “mesenchymal stromal cells”, which should be in fact called “mesenchymal stem/ stromal cells”. Please correct.
  • Line 50-55: The description of IRO is not clear. Please, explain better the two variants of IRO with renal tubular acidosis and CAII deficiency. Which one is manifested by RTA and cerebral calcification? Both or each of them have different manifestation, respectively? Please explain and correct this section.
  • Line 64-65: MSCs (including DP-MSCs from SHED) do not belong to the group of pluripotent SCs. They simply do not fulfill the criteria of pluripotency (e.g. teratoma formation, blastcyst complementation, etc). Marker expression and in vitro differentiation are not sufficient to call SCs to be “pluripotent”. SCs showing characteristics described by Authors may be called e.g. “stem cells exhibiting some features of pluripotent cells” or “pluripotent – like cells”. Please correct.
  • Please also explain or correct the statement “formed-like structures” (line 65). What do Authors really refer to? What is the structure?
  • Line 67-69: Please provide list of references (include particular references) to support the statement that “Studies have shown that stem cells from human exfoliated deciduous teeth (SHED) present an ideal non-controversial and easily accessible cell source for regenerative medicine” There is no references supporting this very strong statement especially that the number of papers regarding DP-MSCs and their regenerative capacity is limited when compared to e.g. BM-MSCs, AT-MSCs or even UC-MSCs. It should more convincing and supported with some citations.
  • The same for the statement in lines 71-73. Please provide references supporting statement about use of DP-MSCs in “varies of medical diseases”.
  • Starting from Introduction section (from line 73), the Authors use the abbreviation “SHED” as a synonym of DP-MSCs, which is a mistake. SHED is not a cell and do not “differentiate” and can not be “reprogrammed”. Please correct in whole manuscript.
  • Line 99-100: Figure 1: The labels describing scales look too small. Please make them larger, especially on both graphs showing cell growth in time (is it time or passage number?)
  • The data on Figure 1 is not well described. It should be better explained in the legend what is shown. It is absolutely unclear. Are these two representative cultures? Two independent isolates? What are these symbols and what they mean? The section 2.1 of results should be also expanded with better description of data.
  • Section 2.2.: Figure 2: Data for “negative” markers along with isotype controls should be shown in Figure 2. The “Note” under the figure (line 113) should be moved to the main text (removed for figure). Figure legend must be expanded. More information must be included for data description including how the “Percentages %” were calculated. What data are shown (which samples)?
  • Section 2.3. There is lack of introduction why this assay was performed. Please explain why the CFU capacity is important and why it was done. The section has only 2 lines! and Figure.
  • I would recommend combining the sections 2.1 and 2.3 in one chapter referring to the proliferation and colony- forming capacity of the DP-MSCs. These are not really chapters.
  • Figure 3A: Again, the data description is unclear. What is shown on bar graphs? The scales are not described. In the legend: please add information if the data comes form one clone? Do the bars show a mean form all clones? Please expand the descriptions.

  • Why the Figure 3 is divided into in fact 2 figures (3A and 3B). These should be two independent figures.
  • Figure 3B: Please, make it Figure 4. It is confusing to be “a part” of Figure 3, since fig. 3A and 3 B shows a completely different aspect of DP-MSC activity.
  • Section 2.4.: Did the Authors verify adipogenic differentiation of DP-MSCs? According to ISCT recommendations, differentiation into 3 mesodermal lineages should be examined for MSCs. Please provide the data and their description.
  • There is no information in the text how the experiments were performed (again the chapter are 4 lines!). Please explain with more details what has been done. Can you conclude based on your data which population differentiated better? Have you performed any semi-quantitative analyses of e.g. differentiated colony numbers, etc.? Please provide better description of this part of the Results.
  • Figure 3B: What are the controls on Figure3B? The images with larger magnifications of differentiated cells must be included into the Figure 3B. There are some abbreviations: “DPC” that are unknown – please correct the labels.
  • Section 2.5: This is the most valuable and novel part of the paper, which is in my opinion completely under described. It should be better explained how the experiment and analysis was done and what are the data. What does “WCL” mean? – please explain.
  • Please also better explain how the comparison analysis was performed. The statement: “This analysis was taken as baseline protein profile to examine how much similarity of difference between DP-MSC derived from OP and HC subjects” (lines: 139-141) is completely unclear. Please correct and clarify for better understanding.
  • What are these 86 proteins significantly different in the expression between HC and OP DP-MSCs? Please include the full list of these proteins (along with the fold difference between both populations) as a Supplementary materials. Arrange the proteins according to the decrease in fold difference OP:HC (the highest values first)
  • The section 2.6. should be combined with section 2.5 into one chapter related to the proteomic analysis of DP-MSCs.
  • Section 2.6.: In this section, please explain if DP-MSCs were differentiated separately into chondrocytes and osteoblasts or it was kind of mixed differentiation (?).
  • What is the difference between the 86 (5.7%) of proteins differently expressed between undifferentiated HC and OP cells (line 146) and the 303 (21.7%) detected between those undifferentiated cells in another analysis (line 156-157). Should it be the same pool of proteins? Please explain.
  • Please include the list of 79 proteins differently expressed between HC and OP DP-MSCs (along with the fold difference between both populations) as a Supplementary materials.
  • Section 2.7: Figure 4 should be in fact a Figure 5. Please correct. Figure 4 in such a horizontal arrangement is difficult to read (is too small). Please arrange into vertical set up. The Figure should occupy at least whole page to be visible and readable. It is not possible to read it now.
  • Table 2: Please explain what the values in the expression fold change OP:HC means. Which proteins do have a higher expression in HC cells? Provide information for the publication Readers how to follow and understand the data. Please explain and comment on the the expression of particular prominent proteins – e.g. MAPK3 (is the expression 25 times higher or lower in OP or in HC cells), FERMT2 (is the expression 6 times higher or lower in OP or in HC cells) and other.

  • Since as mentioned by the Authors, it is a first proteomic analysis of DP-MSCs, the description of the data should be much more comprehensive.
  • There is an information about commercially available line Ax390, which supposed to be also analyzed in this study, but it was never mentioned in the Result section. The Authors have to explain exactly which data was obtained based on primary DP-MSC clones and which ones based on the cell line. It must be clearly indicated which data is obtained based on what material.
  • Line 207-208: The statement should be rewritten and better explained. It is unclear now. The Authors should also explain how the findings in Ref. 19 is similar to the findings in this study. It is unclear.
  • Line 210-213: Provide information if the proteins identified as differently expressed in ADO-2-iPSCs and NC-iPSCs (Ref. 20) belonged to the same groups/ pathways/ are the same proteins identified as differently expressed between OP and HC DP-MSCs in this study. The fact that similar % of proteins (about 5%) is differently expressed may not necessarily indicate similarities.
  • Line 235-236: The Authors mentioned that gene expression analyses was also performed. However, this data is not included into the manuscript. It is unclear. Please include the mRNA data into the main manuscript or to the Supplemental materials or specify clearly that the it was “protein expression”. The “gene expression” statement indicates mRNA data rather then protein.
  • The Discussion should be significantly polished in terms of language. The sentence in lines 243-245 is completely unclear. Please correct.
  • Line 246-248: Explain what it means. In which cells the expression of ColA1 was higher? How does it matter here? The same, the statement in lines 249-252 should be better explain to provide the meaning for this data.
  • There is no conclusion at the end of the Discussion section, which would be appreciated. The Authors should add a short sum up and conclusion of their study.
  • Section 4.3. Cell Proliferation Assay is very broadly described. Please provide the details of this assay.
  • Line 312-314: Please explain better. It is unclear how the 2D gel electrophoresis was used to verify the cell quality. Actually, flow cytometry analysis confirming stable antigen expression should be used to verify stability and purity of the MSC phenotype in the culture and to identify the appropriate passage for cells to be used. What was the antigen expression on passage 1, 2, 3 and 4? Did Authors check this?
  • Section 4.5. Please provide the list of antibodies used in the study (clones, colors, vendor). The table would be most suitable.
  • Lines 395-398: It is too strong conclusion based on the data included into this manuscript. It should be more objective. Please correct. The same line: 404-407 – please correct. Much more extended studies would need to be performed to conclude that DP-MSCs may be banked and used instead of allogeneic SC therapy, which has been employed in the clinic and used in patients for last dozen years.

Minor changes:

  • Line 103-105: There is no reference for ISCT position paper. Please, correct.
  • Lines 226-228: Two different abbreviations of dental pulp MSCs are used. Please correct and use the only one abbreviation.
  • Line: 318, 323 and further in Material and Methods: Correct the letter in numbers to superscript.
  • Line: 322: What does “Axol” mean?
  • Figure 5 and the Workflow would be more appropriate to place at the beginning of the section of Material and Methods for better understanding of the following methodology.
  • Section 4.7. What medium was used. Please provide some details.
  • Line 399: Please correct ”immun-ophenotyping”.

Reviewer 2 Report

The manuscript of Alkhayal et al. titled “Proteomic Profiling of the First Human Dental Pulp 2 Stem Cells from Carbonic Anhydrase II Deficiency 3 Osteopetrosis Patients” compare dental pulp stem cells from healthy donors vs dental pulp isolated from osteopetrosis patients. It is not clear the aim of the study and what the authors would like to demonstrate. As reported in the introduction section “Osteopetrosis comprises a group of rare 34 heterogeneous hereditary disorders characterized by defective osteoclast activity where bones 35 become sclerotic, thick, weak and brittle”, this disease affects osteoclasts that derive from monocytes-macrophages, while dental pulp stem cells represent a mesenchymal stem cells from the neural crest. What is the link between the osteopetrosis and teeth and dental pulp stem cells? Why osteopetrosis shoud influence the biological properties of dental pulp stem cells? 

Moreover, materials and methods are poor, immunofluorescence, western blot, PCR analyses must be performed in order to provide quantitative data.